# Effects of Testosterone Replacement Therapy on Metabolic Syndrome in Male Patients-Systematic Review

**DOI:** 10.3390/ijms252212221

**Published:** 2024-11-14

**Authors:** Nicola Mlynarz, Miłosz Miedziaszczyk, Barbara Wieckowska, Edyta Szalek, Katarzyna Lacka

**Affiliations:** 1Student Research Group, Endocrinology Section at the Department of Endocrinology, Metabolism and Internal Medicine, Poznan University of Medical Sciences, 60-355 Poznan, Poland; nicolamlynarz@gmail.com; 2Department of General and Transplant Surgery, Poznan University of Medical Sciences, 60-352 Poznan, Poland; 3Department of Clinical Pharmacy and Biopharmacy, Poznan University of Medical Sciences, 60-806 Poznan, Poland; szalekedyta@wp.pl; 4Department of Computer Sciences and Statistics, Poznan University of Medical Sciences, 61-701 Poznan, Poland; bbwieckowska@gmail.com; 5Department of Endocrinology, Metabolism and Internal Medicine, Poznan University of Medical Science, 61-701 Poznan, Poland

**Keywords:** metabolic syndrome, male, testosterone therapy, testosterone, obesity

## Abstract

Metabolic syndrome (MS) comprises several symptoms or disorders that significantly increase the risk of developing atherosclerosis and type 2 diabetes. This study aims to determine the direct impact of testosterone therapy on the components of MS; although excluding type 2 diabetes cases. The authors conducted a systematic literature search of PubMed, Scopus, and Cochrane databases without date limits, using keywords such as “testosterone therapy”, “metabolic syndrome” and “men”. The studies included in our review focused on the effects of testosterone replacement therapy (TRT) in male patients with MS, yet rejecting individuals where type 2 diabetes constituted the only diagnosis. A meta-analysis was performed using PQStat v1.8.6 software. The overall effect size (mean difference) was calculated using a random effects model. Our meta-analysis indicates that testosterone therapy leads to improvement in the components of MS. Significant reductions were observed in waist circumference (WC) (95% CI: −0.709 to 0.094; *p* = 0.011), as well as in triglycerides (TG) (95% CI: −0.474 to 0.120; *p* = 0.039). These findings support the potential therapeutic benefits of testosterone treatment in managing MS. However, further research is vital to explore the long-term effects and the safety of this therapy in patients with metabolic syndrome.

## 1. Introduction

The term “metabolic syndrome” (MS) was introduced in 1981 by German researchers, Hanefeld and Leonhardt, who also developed the criteria for diagnosing MS, which included the coexistence of obesity, diabetes, hyperlipidaemia, arterial hypertension and gout [1].

Another milestone was the phrase “syndrome X”, coined in 1988 by Gerald Reaven. He described it as an increased risk of cardiovascular disorders, if the patient suffered from insulin resistance, compensatory hyperinsulinemia, varying degrees of glucose tolerance, increased plasma triglycerides (TG) concentration and decreased plasma high-density lipoprotein (HDL) cholesterol. Subsequently, the term “syndrome X” gradually transformed into the insulin resistance syndrome [2].

Although initially the diagnostic basis for MS constantly changed, in 1998, the World Health Organization (WHO, World Health Organization) recognized that the most significant criterion for diagnosing the disease was insulin resistance coupled with at least two out of four factors simultaneously present: central obesity, dyslipidaemia, high blood pressure and microalbuminuria [3]. In 2004, the International Diabetes Federation (IDF) initiated a workshop to standardize the diagnostic requirements of MS. The expert panel included representatives of the WHO, The European Group for the Study of Insulin Resistance (EGIR) and the National Cholesterol Education Program—Third Adult Treatment Panel (NCEP ATP III).

According to WHO guidelines, criteria were to involve: insulin resistance/impaired glucose tolerance and at least two additional factors (hypertension ≥ 140/90 mmHg, hypertriglyceridaemia ≥ 150 mg/dL, HDL cholesterol in male (M) patients < 35 mg/dL and in female (F) patients < 39 mg/dL, waist to hip ratio in M > 0.9 and in F > 0.85 and/or body mass index (BMI) > 30, urinary albumin excretion rate ≥ 20 qg/min, or albumin: creatinine ratio ≥ 30 mg/g. As an additional diagnostic criterion, EGIR recommended a fasting glycaemia> 110 mg/dL, although without diagnosed diabetes, as well as rejected microalbuminuria as a factor. The other criteria were concordantly accepted, with minor changes in their threshold values. In contrast, according to NCEP ATP III, insulin resistance or glucose intolerance were not considered necessary for diagnosing MS and, similarly to EGIR, they dismissed microalbuminuria [4].

The authors of the presented paper support the diagnostic criteria outlined by the IDF, which consider central obesity mandatory, with WC ≥94 cm in M and ≥80 cm in F in Europe, and additionally include two of the four concomitant factors: elevated TG ≥ 150 mg/dL or specific treatment for this lipid disorder, reduced HDL cholesterol <40 mg/dL in M or <50 mg/dL in F or specific treatment for this lipid disorder, increased blood pressure: systolic BP ≥ 130 or diastolic BP ≥ 85 mmHg, or treatment of the previously diagnosed hypertension and elevated fasting glycaemia ≥ 100 mg/dL, or previously diagnosed type 2 diabetes [5]. These data are presented in Table 1.

In order to diagnose the metabolic syndrome, it is essential to identify markers of predictive significance. Such markers include chemerin, which is a pro-inflammatory adipokine recognized as a component of pathogenesis MS, a predictor of diabetes and early atherosclerosis [6]. Furthermore, studies confirm the presence of insulin resistance, low levels of total testosterone and decreased levels of sex hormone binding globulin (SHBG) in MS cases, particularly in older men [7]. The research conducted by Comacho et al. indicated that weight gain was associated with a proportional decrease in both testosterone and SHBG levels. Conversely, weight loss was correlated with an increase in these parameters [8].

Testosterone replacement therapy (TRT) has been successfully applied for 70 years, however, it was not until the 1990s that formulations were introduced to provide physiological serum testosterone levels. In order to provide a whole-body testosterone effect, TRT necessitates the use of natural testosterone (T). This, in turn, requires aromatization to oestradiol and a subsequent reduction to 5-alpha dihydrotestosterone (DHT)—these processes occur at physiological rates [9].

Although TRT is hardly a new treatment modality, there are very few studies investigating the impact of the therapy on the individual components of the metabolic syndrome, particularly in cases where there are no comorbidities. The sources include a 30-week study involving 184 male patients from Moscow [10] and a 1-year Japanese study with 65 male participants [11].

Our review comprises studies in which patients were administered testosterone intramuscularly (i.m.). Absorption i.m. increases with the length of the chain of esterified fatty acids at the 17β position. In addition, the site and the amount injected, as well as the oily carrier, are crucial. Testosterone undecanoate (TU) shows the longest carbon side chain, containing 11 carbon atoms, compared to 8 found in testosterone enenthate (TE). In the case of TU, testosterone levels are measured prior to each successive injection, and TE—one week after receiving the dose [9]. In the available scientific literature, the results regarding the effectiveness of testosterone replacement supply remain inconclusive. Thus, the aim of our study was to summarize the available data with regard to the effect of TRT on the components of the MS in men.

## 2. Results

### 2.1. Glycaemia

In terms of fasting glycaemia prior to testosterone treatment, the study group was compared with the control group, and the comparison showed no statistical significance, yet higher values were observed in the study group. However, for glycaemia, the overall effect was almost zero. Figure 1 shows the effect of testosterone on fasting glycaemia.

Studies with sufficient data on the effect of testosterone on FG demonstrated a non-significant decrease in glycaemia. The standardized mean difference in the random model was −0.197 mmol (mM) (95% CI: −0.428 to 0.331; *p* = 0.093). The heterogeneity between the studies was I2 = 0.00% (95% Cl for I2 = 0.00 to 0.00; *p* = 0.9767). The results of the Egger’s test, i.e., 0.0717 (*p* = 0.0001), indicated there was a statistically significant publication load.

### 2.2. Cholesterol

A similar overall effect was observed after treatment with regard to cholesterol levels, which were higher prior to treatment in the control group. Figure 2 shows the effect of testosterone on cholesterol.

Studies with sufficient data regarding the effect of testosterone on cholesterol demonstrated a non-significant decrease in cholesterol. The standardized mean difference in the random model was −0.110 mM (95% CI: −0.341 to 0.120; *p* = 0.346). The heterogeneity between studies was I2 = 0.00% (95% Cl for I2 = 0.00 to 0.00; *p* = 0.4087). The results of the Egger’s test, i.e., 2.0286 (*p* = 0.0001), showed a statistically significant publication load.

### 2.3. Triglycerides

Different results were observed for TG. Before treatment, the overall effect was almost zero; however, the study group showed lower TG levels compared to the control group following the treatment. Figure 3 shows the effect of testosterone on triglycerides.

Studies with sufficient data investigating the effect of testosterone on TG demonstrated a significant decrease in TG. The standardized mean difference in the random model was −0.243 mM (95% CI: −0.474 to 0.127; *p* = 0.039). The heterogeneity between studies was I2 = 0.00% (95% Cl for I2 = 0.00 to 0.00; *p* = 09872). The results of the Egger’s test—0.0395 (*p* = 0.0001)—showed a statistically significant publication load.

### 2.4. High Density Lipoprotein

Before the treatment onset, HDL levels were higher in the study group as compared to the control group, whereas after the treatment, a more favourable overall effect was found in the study group. Figure 4 shows the effect of testosterone on high density lipoprotein.

Studies with sufficient data concerning the effect of testosterone on HDL demonstrated a non-significant increase in HDL. The standardized mean difference in the random model was 0.103 mM (95% CI: −0.269 to 0.475; *p* = 0.587). The heterogeneity between studies was I2 = 49.12% (95% Cl for I2 = 0.00 to 0.00; *p* = 0.1609). The results of Egger’s test, which amounted to −3.4472 (*p* = 0.0001), showed a statistically significant publication load.

### 2.5. Waist Circumference

There were no statistically significant differences in WC in both the study and control groups before the initiation of the treatment. However, the overall effect following the treatment was more beneficial in the control group where higher values were observed in comparison with the study group. Figure 5 shows the effect of testosterone on waist circumference.

Studies with sufficient data investigating the effect of testosterone on WC demonstrated a significant decrease in WC. The standardized mean difference in the random model was −0.402 centimetres (cm) (95% CI: −0.709 to 0.094; *p* = 0.011). The heterogeneity between studies was I2 = 30.34% (95% Cl for I2 = 0.00 to 0.00; *p* = 0.2308). The results of the Egger’s test. i.e., 2.9952 (*p* = 0.0001), demonstrated that there was a statistically significant publication load.

## 3. Discussion

In most meta-analyses, researchers address MS coupled with other concomitant diseases, since the studied patients frequently suffer from other comorbidities. Therefore, it is impossible to determine the true impact of testosterone on lipid metabolism, glycaemia, WC or blood pressure only in terms of MS [12,13,14,15]. To the best of our knowledge, the presented paper constitutes the first meta-analysis to report the results of testosterone replacement solely in MS.

There is a variety of methods involved in the treatment of the metabolic syndrome, although they are not always sufficient. Table 2 outlines the current treatment options for MS, including lifestyle modifications, pharmacological interventions, hormone therapy, bariatric surgery, and behavioural therapy [16,17].

Testosterone and DHT (a product of testosterone metabolism by 5α-reductase) inhibit adipogenesis through the androgen receptor-mediated nuclear translocation of β-catenin and the activation of downstream Wingless-related integration site (Wnt) signalling [18]. DHT blocks the synthesis of subcutaneous fat, whereas oestradiol accounts for blocking the growth of visceral fat tissue [19]. However, a clear reduction in obesity in male and hypogonadal mice is observed only in the case of testosterone replacement.

Notably, testosterone affects carbohydrate and lipid metabolism, bones, blood pressure, muscle mass, obesity and cognitive disorders [20]. The effects of its deficiency are presented in Figure 6.

Testosterone and its metabolites, DHT and oestradiol, play different roles in regulating total fat mass and fat distribution. Hence, investigating which of these parameters and to what extent regulates the occurrence of obesity could prove useful in therapeutic decisions regarding the substitute administration of these substances [19].

### 3.1. Triglycerides

Testosterone physiologically affects male body composition by means of inhibiting adipogenesis and stimulating myogenesis, as well as by playing a role in the metabolism of carbohydrates, lipids and proteins [21]. Decreased tissue testosterone levels stimulate pluripotent stem cells to mature and facilitate triglyceride storage in adipocytes, resulting in an increased adipocyte mass, which contributes to greater insulin resistance [22]. Additionally, it has been established that weight loss increases testosterone levels in middle-aged and older male [8]. As a result, this suggests a bidirectional relationship between testosterone levels and body weight.

Furthermore, adipose tissue is developed by excessive consumption of carbohydrates and fats without adequate physical exercise. This stems from an increase in free fatty acids in the blood, which are subsequently stored in adipocytes. In the event of impaired lipid oxidation in the liver, lipogenesis occurs and further increases in blood lipids [22].

### 3.2. Waist Ircumference

According to the results we compiled, a significant reduction in WC was observed (−0.402 cm; *p* = 0.011) was observed following testosterone treatment. It is also worth bearing in mind that increased WC accounts for a higher incidence of hypertension [23], the complications of which include stroke, kidney failure, cardiovascular disease, such as myocardial infarction, cardiac hypertrophy, and heart failure [24]. In fact, an increased WC may result from central obesity, which, combined with genetic factors and lack of exercise, contributes to the development of insulin resistance. This condition, in turn, promotes the development of hypertriglyceridaemia, hypertension, and endothelial dysfunction [22].

Patients with obesity are leptin-resistant, which, in addition to inhibiting the hypothalamic-pituitary-testicular response to decreasing androgen levels, also directly inhibits testosterone production by suppressing the stimulation of Leydig cells by means of gonadotropins secreted in the arcuate nucleus of the hypothalamus [22,25]. This stems from impaired mitochondrial function and compromised testosterone synthesis due to inhibition of enzymes and proteins caused by the accumulation of oxidized low-density lipoproteins [26].

In our study, we found a positive effect on reducing glycaemia (−0.197 mM, *p* = 0.093), total cholesterol (−0.111 mM; *p* = 0.345), TG (−0.243 mM; *p* = 0.039) and WC (-0.402 cm; *p* = 0.011) and an increase in HDL (0.121 mM; *p* = 0.486). Each component of the MS separately increases the risk of cardiovascular diseases, such as myocardial infarction, heart failure, atherosclerosis, and microcirculation disorders [25]. Its beneficial effect on parameters such as glycaemia, TG, or HDL is validated by the reduction in cardiovascular mortality [27,28]. Figure 7 presents the effects of testosterone therapy on the components of MS.

Notably, testosterone therapy also increases bone mineral density, increases libido, and also has a cardioreceptive effect [9]. It is also worth bearing in mind that cardiovascular risk is reduced by such factors as improved endothelial function [5]. Testosterone activates endothelial nitric oxide synthase (eNOS), which leads to an increased nitric oxide (NO) production, thereby resulting in the dilation of blood vessels and enhanced circulation [29].

Furthermore, it has been observed that prolonged testosterone therapy results in greater health benefits, which supports the rationale behind extending its supplementation [8,12]. The Moscow study lasted 7.5 months (30 weeks) and the study group received TU at a dose of 1000 mg i.m [10]. In terms of the Japanese study, it lasted 12-months, and the study group received TE at a dose of 250 mg i.m [11]. It was reported that patients treated for a longer period than the target normal testosterone level demonstrated the most significant benefits from the therapy [12]. Hypogonadism in the Moscow study was diagnosed at a total testosterone value of 350 ng/dl, or on the basis of the calculated free testosterone i.e., 65 pg/ml [10], whereas according to the Japanese criteria it was diagnosed when free testosterone (FT) was ≤11.8 pg/ml [11].

Type 2 diabetes is considered one of the criteria of MS. In a study involving 156 men, the results clearly showed that glycaemia levels decreased following testosterone treatment. Moreover, body weight and WC were also lower than before testosterone treatment [14]. A registry study, involving 255 male with hypogonadism, demonstrated that TRT reduced body weight, WC, as well as BMI [15]. In one of the meta-analyses including patients with MS and type 2 diabetes, an improvement in glycaemia and lipid parameters [HDL, LDL (low-density lipoprotein) and TG] and a decrease in body weight and WC were observed [18]. The aforementioned observations are consistent with our results.

Even though testosterone therapy has been used for 70 years [9] and its beneficial effects are widely recognised, it is also vital to address the negative impact of TRT. It accounts for increased haematocrit, which is linked to an elevated risk of thromboembolism and other cardiovascular events. This is due to the fact that the therapy results in an increased iron mobilization, stimulating erythropoiesis and, consequently, increasing haematocrit [30]. In particular, testosterone esters, which were incorporated in the studies included in our meta-analysis, are associated with a higher red blood cell count, and consequently increase the risk of cardiovascular complications [31]. Moreover, TRT inhibits spermatogenesis. Although it is possible to return to the baseline level after the treatment, individuals who are actively trying to conceive offspring should be aware of this effect [32]. It is also crucial that TRT is limited in cancer patients, as the therapy may cause exacerbation of breast or prostate cancer [33]. Nevertheless, certain studies showed no adverse events were reported in the course of TRT, for instance the two-year SETH2 study, which demonstrated a positive effect of TRT on WC, TG and a number of other parameters, such as glycaemia, body weight or BMI [34]. Similarly, in the study by Haider et al., no serious cardiovascular complications or urological problems were reported [35].

The limitations of our study stem from the limited number of studies included in the meta-analysis. Nevertheless, testosterone replacement therapy has gained considerable interest, thus, there is a need for well-designed studies employing this method [36]. The keyword search involved only three terms, which may have contributed to the omission of certain papers and, thus, their exclusion from our meta-analysis. Furthermore, it is essential to conduct studies which compare different dosages and durations in order to establish standardized treatment protocols. There is also a need for detailed research exploring how TRT affects metabolic pathways at a molecular level. Additionally, conducting comprehensive genetic testing may identify polymorphisms in genes related to testosterone metabolism, androgen receptor sensitivity, and metabolic pathways. Further research into the impact of TRT in patients with MS also seems warranted.

## 4. Materials and Methods

### 4.1. Study Design

Two authors independently conducted a systematic literature search of Scopus (1972–2024), PubMed (1976–2024), and Cochrane (1998–2024) databases, using key words, such as “testosterone therapy”, “metabolic syndrome”, and “men”. The review included papers which: focused exclusively on testosterone therapy only in MS in men, were published within the last 48 years, were available in full text, and were published in English.

In fact, the authors initially obtained 718 results, however, 716 were discarded since the papers either described cases where the patients presented with comorbidities, and—therefore—were not relevant to the scope of our study, or testosterone treatment was used for other diseases. Our literature review is presented in Figure 8.

The majority of the available literature included patients with diseases other than MS, or they investigated testosterone treatment effects in patients with both type 2 diabetes and the metabolic syndrome, who were considered a single group. The strict approach we adopted prompted us to reject such studies.

The authors found two papers (identifier: NCT00696748) in the literature based on the same group of patients, although with different aims [10,37]. We decided to include into our meta-analysis a 30-week study of 184 men [10] from Moscow and a 1-year Japanese study of 65 male participants [11]. Study NCT00696748 comprised patients from Moscow, whereas patients with hypogonadism in the Japanese study were selected from the EARTH study. All the participants involved met the criteria for the diagnosis of metabolic syndrome and hypogonadism. In both studies, 145 patients from the study groups received testosterone replacement therapy, and 104 patients from the control groups were administered placebo. The study protocol of our study was registered (ID CRD42024571422).

### 4.2. Bias Assessment

Bias assessment was conducted using the RoB 2 for each individual study, by addressing the signalling questions in order to determine the risk of bias vital to draw conclusions from the accessed data. Two researchers were involved in this process. The sources obtained for bias assessment included research papers and study protocols, if available. The final conclusion was based solely on studies graded “low” or “some concerns”. Figure 9 shows the bias assessment.

A meta-analysis was performed using PQStat v1.8.6 software. The overall effect size (mean difference) was calculated on the basis of a random effects model. Due to the inclusion of only two studies, the assessment of heterogeneity and publication bias was omitted. Egger’s test was used to assess publication bias. Evaluation of heterogeneity was performed using the I2 statistic. The overall effect for each parameter was presented in table along with a 95% Confidence Interval and with forest plots separately. Based on our analyses, the main results from the selected studies were summarized and presented in text and/or tabular form.

### 4.3. Explanation for Conducting a Meta-Analysis

Conducting a meta-analysis based on two studies is possible, although it is not perfect. The list below provides some of the reasons:

Low statistical power: meta-analyses are based on statistics, combining results from numerous studies. Two studies appear insufficient to provide sufficient statistical power to detect a significant effect.

Insufficient precision: the results of a meta-analysis based on two studies show a large standard error, hence, they are less precise.

Risk of random error: with such a small number of studies, there is a high risk that the results of the meta-analysis will be unintentionally biased.

No possibility of verifying heterogeneity: meta-analyses are incorporated in order to assess whether research results are consistent. Two studies are not enough to evaluate heterogeneity.

Limited generalizability: the results of a meta-analysis based on two studies may preclude their applicability to other populations, or contexts.

Therefore, meta-analyses involving only two studies are rare. Nevertheless, a two-study meta-analysis may be justified in cases when the purpose of the meta-analysis is only to investigate the direction of the effect rather than its magnitude. Thus, we decided to conduct our meta-analysis.

## 5. Conclusions

Testosterone treatment results in a significant reduction in WC and blood triglyceride levels. Therefore, understanding the underlying mechanisms may be particularly valuable for developing more targeted therapies, as well as for identifying potential biomarkers for the treatment response. The abovementioned data could be instrumental in predicting which patients are likely to respond favourably to TRT, and which individuals may be at an increased risk of side effects.

## Figures and Tables

**Figure 1 ijms-25-12221-f001:**
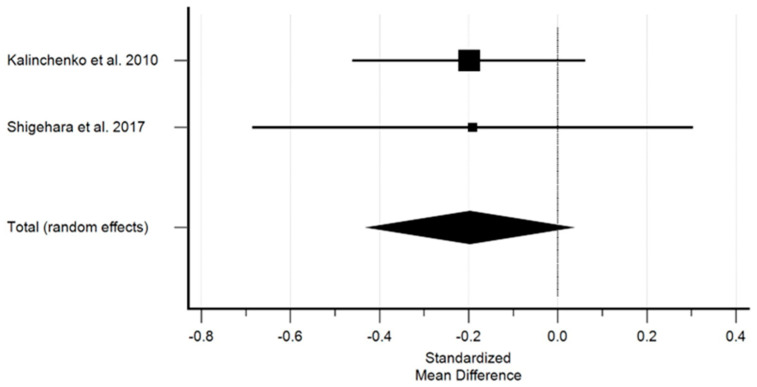
The effect of testosterone on fasting glucose level [10,11]. Legend: The horizontal lines denote the 95% CI, the Square (■) shows the point estimate (the size of the square corresponds to its weight); the diamond shows (♦) the combined overall effects (random effects) of testosterone.

**Figure 2 ijms-25-12221-f002:**
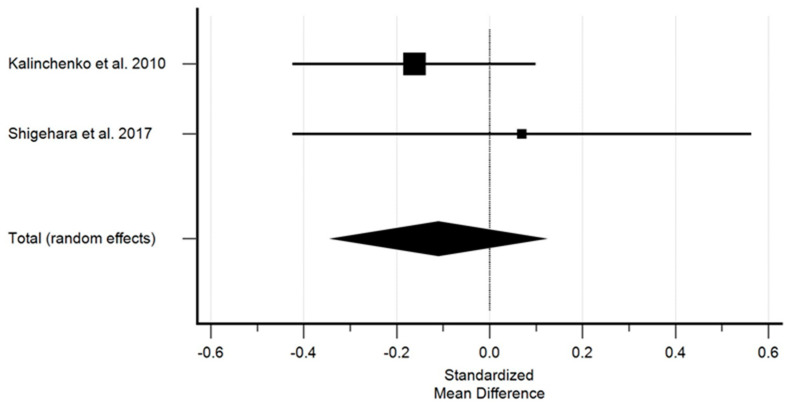
The effect of testosterone on cholesterol [10,11]. Legend: The horizontal lines denote the 95% CI, the Square (■) shows the point estimate (the size of the square corresponds to its weight); the diamond shows (♦) the combined overall effects (random effects) of testosterone.

**Figure 3 ijms-25-12221-f003:**
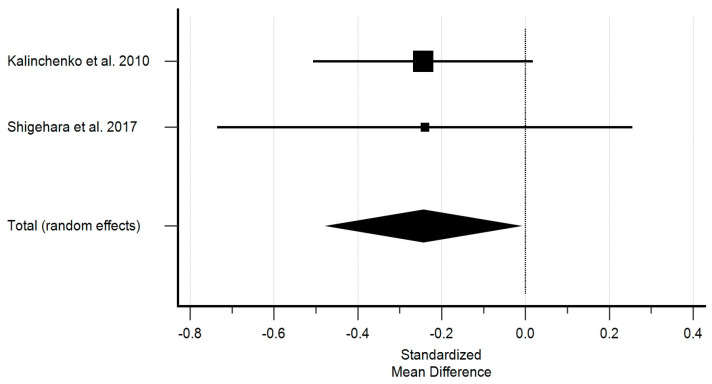
The effect of testosterone on triglycerides [10,11]. Legend: The horizontal lines denote the 95% CI, the Square (■) shows the point estimate (the size of the square corresponds to its weight); the diamond shows (♦) the combined overall effects (random effects) of testosterone.

**Figure 4 ijms-25-12221-f004:**
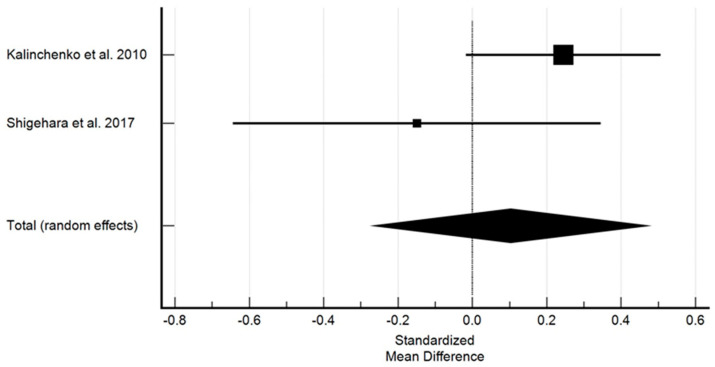
The effect of testosterone on high density lipoprotein [10,11]. Legend: The horizontal lines denote the 95% CI, the Square (■) shows the point estimate (the size of the square corresponds to its weight); the diamond shows (♦) the combined overall effects (random effects) of testosterone.

**Figure 5 ijms-25-12221-f005:**
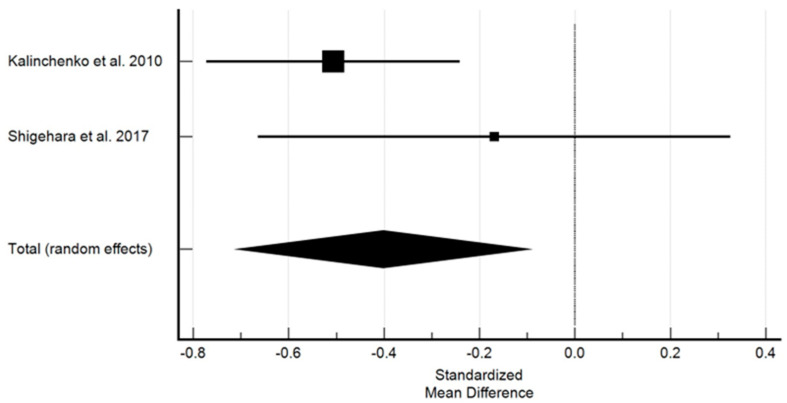
The effect of testosterone on waist circumference [10,11]. Legend: The horizontal lines denote the 95% CI, the Square (■) shows the point estimate (the size of the square corresponds to its weight); the diamond shows (♦) the combined overall effects (random effects) of testosterone.

**Figure 6 ijms-25-12221-f006:**
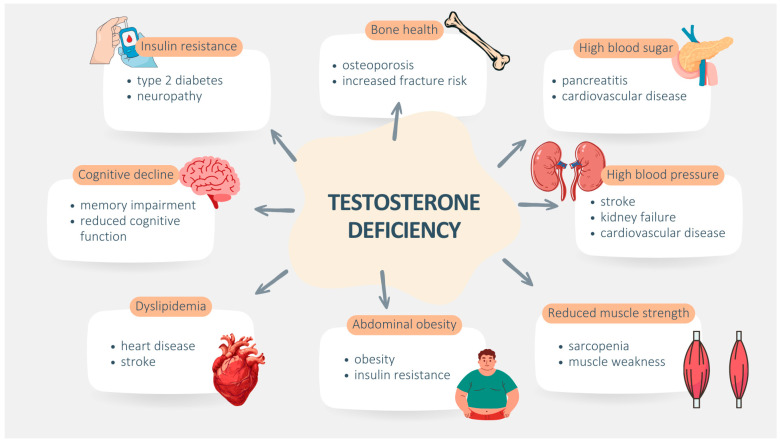
Comprehensive impact of testosterone deficiency on health components and the related issues. This figure presents the extensive effects of testosterone deficiency on various health components. The central circle represents testosterone deficiency, surrounded by key components of MS and other health issues.

**Figure 7 ijms-25-12221-f007:**
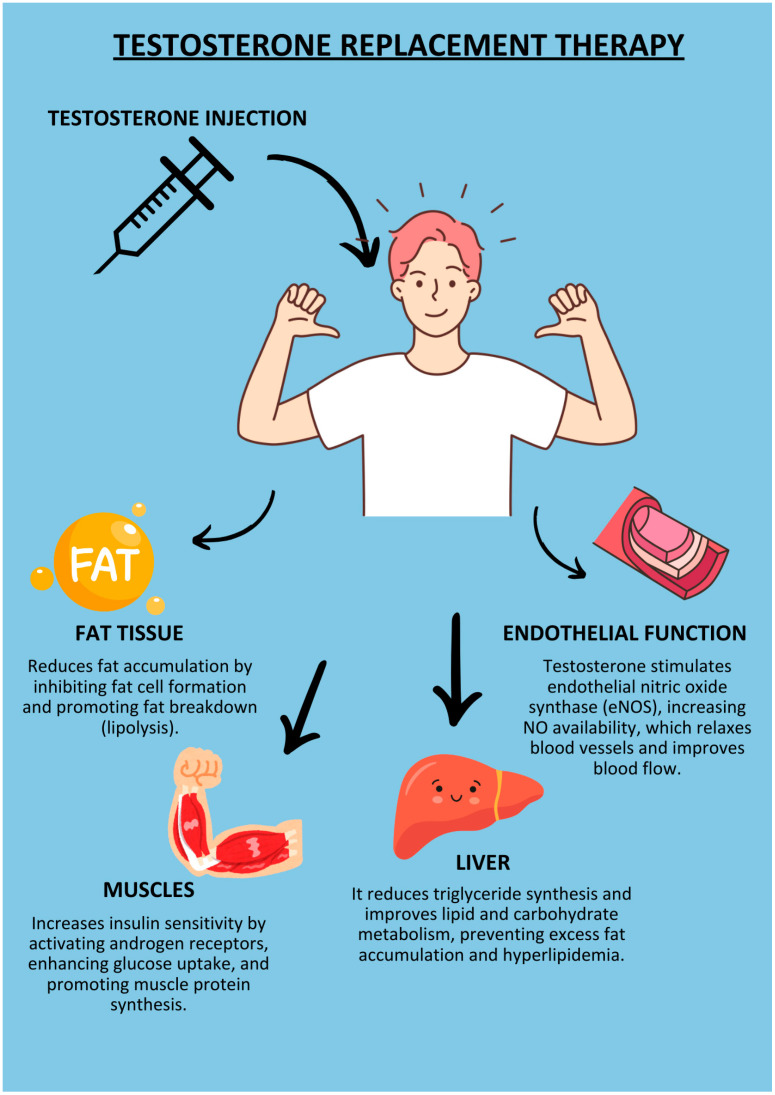
The effects of testosterone replacement therapy on the components of the metabolic syndrome.

**Figure 8 ijms-25-12221-f008:**
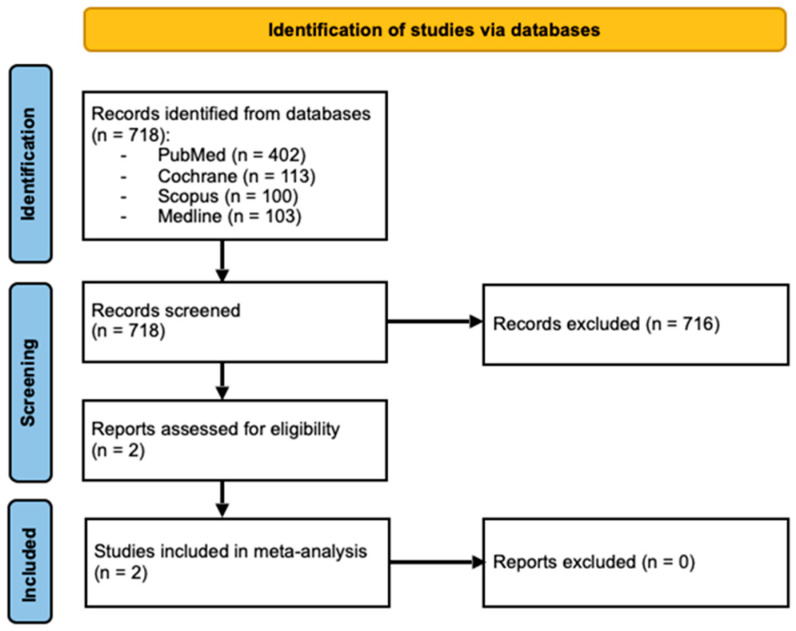
Flow chart summarizing the screening strategy for studies included in the meta-analysis. The chart shows the number of records identified, screened, excluded, and the final studies included in the analysis.

**Figure 9 ijms-25-12221-f009:**
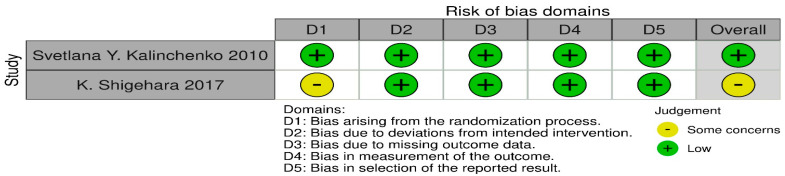
Bias assessment for testosterone replacement therapy [10,11].

**Table 1 ijms-25-12221-t001:** Diagnostic Criteria for the Metabolic Syndrome According to the International Diabetes Federation. WC—waist circumference; TG—triglycerides; HDL—high-density lipoprotein; BP—blood pressure; FG—fasting glucose; M—men; F—women.

Criterion	Description	Thresholds
Central Obesity	Mandatory; measured by WC.	M: ≥94 cm, F: ≥80 cm
TG	Elevated levels or specific treatment for this lipid disorder.	≥150 mg/dL
HDL	Lower levels of HDL or specific treatment for this lipid disorder.	M: < 40 mg/dL, F: < 50 mg/dL
BP	Elevated systolic or diastolic blood pressure, or treatment for previously diagnosed hypertension.	Systolic BP ≥ 130 mmHg, Diastolic BP ≥ 85 mmHg
FG	Elevated fasting glucose levels or previously diagnosed type 2 diabetes.	≥100 mg/dL

**Table 2 ijms-25-12221-t002:** The current methods of metabolic syndrome treatment. The following table summarizes the up-to-date treatment methods for MS, including lifestyle modifications, pharmacological interventions, hormonal treatments, bariatric surgery, and behavioural therapy. Each category has been detailed with specific treatments and their descriptions [16,17].

Category	Treatment	Details
Lifestyle Modifications	Diet	Emphasis on a balanced diet, reduced intake of refined sugars and saturated fats, increased fibre consumption.
	Exercise	Regular physical activity, including aerobic exercises and resistance training.
	Weight Loss	Achieving and maintaining a healthy weight through diet and exercise.
Pharmacological Interventions	Antihypertensives	Medications to control blood pressure.
	Lipid-Lowering Agents	Statins and other drugs to manage cholesterol levels.
	Antidiabetic Medications	Metformin and other medications to control blood glucose levels.
Hormonal Treatments	Testosterone Therapy	Used to treat hypogonadism and potentially improve metabolic parameters.
Bariatric Surgery	Indications	For patients with severe obesity where lifestyle and pharmacological treatments are insufficient.
	Types	Gastric bypass, sleeve gastrectomy, and other surgical interventions.
Behavioural Therapy	Counselling	Supporting lifestyle changes, stress management, and adherence to treatment plans.
	Support Groups	Participation in group therapy to promote sustained lifestyle changes.

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
