# Peer review of "Effects of Testosterone Replacement Therapy on Metabolic Syndrome in Male Patients-Systematic Review"

_ijms, 2024, doi:10.3390/ijms252212221_

Round 1
Reviewer 1 Report
Comments and Suggestions for Authors
In the present work, Młynarz try to review the effects of testosterone replacement therapy on metabolic syndrome in male patients. The manuscript suggested that testosterone therapy has demonstrated positive effects on the components of metabolic syndrome, such as reductions in metabolic syndrome and triglycerides, which supports the potential therapeutic benefits of testosterone treatment in managing metabolic syndrome. However, there are some questions that should be explained.
Major concerns
1. It is known that testosterone therapy has positive effects on men. However, the negative effects on the men for testosterone therapy also should be paid attention.
2. As a review, some fine figures are needed. For example, in Figure 6, some cartoon pictures, including heart, kidney…. In addition, a fine figure for the mechanism of testosterone therapy on metabolic syndrome is also needed.
3. English grammar and writing style should be checked and revised throughout the manuscript. I suggest that it is supported by a professional English language proofreading service.
Minor concerns
1. The writing style of Abstract section is not suitable for IJMS.
2. Keywords are only three, which should be added.
3. Introduction section should be rewritten. Introduction section is too long, and some contents could be moved to other section. In addition, the hypothesis and aim of this study should be added in the end of Introduction section.
4. Figure 1 should be in one page.
5. Lines 206-217, The serial numbers (1., 2. ….) are not right. 2.4.1, 2.4.2…. may be used.
6. Lines 284-285, too short paragraph. Estradiol is a form of estrogen, a female sex hormone. Therefore, this short paragraph should be deleted.
Comments on the Quality of English LanguageExtensive editing of English language required.
Author Response
Dear Reviewer,
We thank you very much for the effort put into the review and for valuable comments. The comments helped us significantly improve the article.
Yours Sincerely,
Authors

Reviewer 2 Report
Comments and Suggestions for Authors
Testosterone replacement therapy (TRT) is the primary treatment for male hypogonadism. It has been confirmed that TRT can improve the symptoms of hypogonadism; however, the metabolic effects of this treatment on male hypogonadism remain controversial. Evidence supporting TRT improvement of glucose control, lipid profile, and weight control is still insufficient. For this reason, screening for testosterone deficiency and TRT for men with T2DM and MetS is not routinely recommended in some countries. In recent years, a large body of evidence has consistently shown that TRT can improve metabolic disorders in MetS, which is a compelling argument to conduct this meta-analysis.
However, some key issues need to be addressed.
1. Were all recommendations included in the PRISMA (Preferred Reporting Items for Systemic Reviews and Meta-Analyses) protocol followed?
2. Although the authors applied rigorous inclusion and exclusion criteria and selected only 2 studies for meta-analysis, testosterone therapy in one case lasted 30 weeks (Kalinchenko et al. 2010), and in the other 12 months (Shigehara et al. 2018). The doses were also different. Do the authors think this is not a limitation of this study?
3. The discussion should be improved. As I understand, the primary goal of this meta-analysis was to determine the efficacy of testosterone treatment in men with MetS. Please focus on the most important observations, as demonstrated by this analysis.
Minor comments
1. Table 1 is hard to read. Please separate the categories and sign the table at the top.
2. Line 70 – should be male/female. Please correct men to male.
3. Line 234 – please move “Figure 1. Effect of testosterone on fasting glucose” under the figure.
4. Line 287 – please explain the abbreviation Wnt.
5. Line 343 – should be: type 2 diabetes.
Comments on the Quality of English LanguageMinor editing of English language required.
Author Response

(The authors gave the same response as above.)

Round 2
Reviewer 1 Report
Comments and Suggestions for Authors
Thanks for author’s responses. However, the writing style still should be checked and revised throughout the manuscript.
1. There are some short paragraphs in Introduction, Materials and Methods sections, and some should be incorporated.
2. Page 2, ‘The European Male Aging Study’ should be revised.
3. For all Figures, figure legends are needed, which are not in italic.
4. Results and discussion section should be divided to two sections (Results and Discussion sections).
5. Figure 8, Colors should be used for the tissues, and the style of calligraphy should be based on the required by this Journal.
6. Figure 9, three tetragons indicate what?
7. doi: (Ref. 30) or https://doi.org/?
The doi for Ref. 3 and 7 are not right. Please check this throughout.
Comments on the Quality of English LanguageExtensive editing of English language required.
Author Response
Dear Reviewer,
The authors thank you very much for the effort put into the review and for valuable comments. The comments helped us significantly improve the manuscript.
We have made corrections to the manuscript based on your comments. Each comment was considered individually and we have made appropriate changes to our manuscript. The manuscript has been revised by an English language expert, and the relevant document confirming the review and proofreading is attached as an additional document.
All answers are presented in the file below with our answers. Once again, we sincerely thank you for your contribution, which allowed us to improve the quality of our manuscript.
Yours Sincerely,
The authors
